

# Development of an LSTM-Broadcasting deep-learning framework for regional air pollution forecast improvement

Haochen Sun[1,2], Jimmy C.H. Fung[1,3,4], Yiang Chen[3], Zhenning Li[3], Dehao Yuan[5], Wanying Chen[3], and Xingcheng Lu[3,4]

[1]Department of Mathematics, Hong Kong University of Science and Technology, Clear Water Bay, Hong Kong SAR, China

[2]Department of Computer Science and Engineering, Hong Kong University of Science and Technology, Clear Water Bay, Hong Kong SAR, China

[3]Division of Environment and Sustainability, Hong Kong University of Science and Technology, Clear Water Bay, Hong Kong SAR, China

[4]Atmospheric Research Center, HKUST Fok Ying Tung Research Institute, Guangzhou, China

[5]Department of Computer Science, University of Maryland, College Park, Maryland, USA

*Correspondence to*: Xingcheng Lu (xingchenglu2011@gmail.com)

**Abstract.** Deep-learning frameworks can effectively forecast the air pollution data for individual stations by decoding time-series data. However, most of the existing time-series-based deep-learning models use offline spatial interpolation strategies

and thus cannot reliably project the station-based forecast to the spatial region of interest. In this study, the station-based long short-term memory (LSTM) technique was extended for spatial air quality forecasting by combining a novel deep-learning layer termed the broadcasting layer, which incorporates a learnable weight decay parameter designed for point-to-area extension. Unlike most existing deep-learning-based methods that isolate the interpolation from the model training process, the proposed end-to-end LSTM-broadcasting framework can consider the temporal characteristics of the time series and spatial

relationships among different stations. To validate the proposed deep-learning framework, $PM_{2.5}$ and $O_3$ forecasts for the next 48 h were obtained using 3D chemical transport model simulation results and ground observation data as the inputs. The root mean square error associated with the proposed framework was 40% and 20% lower than those of the Weather Research Forecast–Community Multiscale Air Quality model and an offline combination of the deep-learning and spatial interpolation methods, respectively. The novel LSTM-broadcasting framework can be extended for air pollution forecasting in other regions

of interest.

## 1 Introduction

Aggravated by industrialization and economic development, air pollution has received increasing attention in recent years. Fine suspended particulate matter ($PM_{2.5}$) and ozone ($O_3$), as prominent secondary air pollutants, can adversely influence





human health and society (e.g., deteriorated visibility may lead to traffic delays). Accurately forecasting the levels of these
two pollutants at the regional scale can provide the information necessary for relevant parties and the general public to address
the threats posed by air pollution and implement appropriate counteractive measures (e.g., emission reduction or curtailment
of unnecessary outdoor activities). To this end, several forecasting models have been developed. Three-dimensional (3D)
numerical models have been applied worldwide to obtain regional forecasts of air pollution levels. Based on historical emission
inventories and physical or chemical parameterization schemes, these numerical models simulate the formation, transmission,
and destruction of air pollutants and forecast the regional air quality over a long prediction horizon (e.g., 120 h). However, the
forecasts provided by such numerical models are prone to significant errors owing to the uncertainty and hysteresis of the
emission inventories and bias in the simplified parameterization schemes and meteorological simulations (Gilliam et al., 2015;
Holnicki & Nahorski, 2015; Tang et al., 2009).

In recent years, machine learning algorithms have been widely applied to predict air quality (Janarthanan et al., 2021; Mao et
al., 2021; Samal et al., 2021; Wu & Lin, 2019; Kim et al., 2019). As the future air quality is correlated with historical values,
ground observations can be input to machine learning models to obtain forecasts. The forecasting process can be formulated
as a time-series task, with the input and training targets being hourly ground observations. Most studies (Ayturan et al., 2018;
Huang & Kuo, 2018; Tsai et al., 2018; Zhao et al., 2020) have applied long short-term memory (LSTM, Hochreiter et al., 1997)
frameworks—a variant of recurrent neural networks (RNNs) and a state-of-the-art deep-learning technique—to accomplish
the time-series tasks. Different LSTM frameworks (or other variants of RNNs) can be applied for different time-series tasks.
For example, if the output temporally post-dates the input, LSTM encoder–decoders (Sutskever et al., 2014) can be applied.
In contrast, if the output and input are in the same temporal domain, bidirectional LSTMs (Schuster & Paliwal, 1997) can be
used. However, because the air quality depends on many factors other than historical values, the correlation between the future
air pollution conditions and past ground observations is weak, especially in the case of large time lags, and the effective
prediction horizon is constrained, typically to no more than 24 h (Bui et al., 2018; Li et al., 2020; Qin et al., 2019). Moreover,
most of the abovementioned studies focused on obtaining accurate forecasts for specific ground monitoring stations, and thus,
deep-learning models that can forecast the air quality on a regional scale are lacking.

Several studies have attempted to develop deep-learning-based models to obtain regional air pollution forecasts by combining
ground observation data and numerical model results through spatial interpolation methods. For example, the LSTM-3D-
variational assimilation (VAR) model (Lu et al., 2021) combines ground observations and 3D numerical models with the
LSTM and 3D-VAR data assimilation techniques. This model can achieve accurate regional forecasts with a prediction horizon
of 24 h; however, substantial computation power is required (one hour of computing time is required to obtain a 24 h forecast
using two AMD EPYC 32-core processors). The LSTM-Weather Research Forecast–Community Multiscale Air Quality
(WRF-CMAQ) model (Sun et al., 2021) combines ground observations and WRF-CMAQ models to achieve highly accurate
regional forecasts with a prediction horizon of 48 h. However, the system requires a customized spatial correction (SC) scheme
(e.g., numerical interpolation methods), and the accuracy at general locations is lower than that at the ground monitoring
stations, the data of which are used for deep-learning model training. Bi et al. (2022) used the random forest algorithm to





calibrate the numerical simulation based on chemical transport models. However, this model relies on interpolation methods (e.g., ordinary kriging). Moreover, the parameters needed for the spatial interpolation schemes are not included in the training

process when constructing the deep-learning framework, and the spatial correlations between different stations cannot be introduced as a constraint (Zhou et al., 2020; Hähnel et al., 2018).

With advances in deep-learning techniques, sophisticated architectures have been developed to incorporate spatial–temporal correlations for regional air pollution forecasting. Pak et al. (2019) developed a spatial–temporal convolutional neural network (CNN)-LSTM network to predict the next day's daily average $PM_{2.5}$ concentration in Beijing, China. Qi et al. (2019) applied

a graph neural network (GNN) to take into account the spatial correlations of multiple ground monitoring stations in the Jing-Jin-Ji region, China, and enhance the forecast accuracy at these stations. Han et al. (2021) proposed a MasterGNN structure to explore the spatial–temporal information and forecast the air quality and weather at a given set of ground monitoring stations. However, the forecasts obtained by these architectures are restricted to a city-wide average or fixed set of ground monitoring stations. Therefore, these models cannot be applied for regional forecasting and predicting the pollutant concentrations at

specific locations.

In this study, to obtain accurate forecasts for a longer period and consider the spatial characteristics, an end-to-end deep-learning model that can forecast the regional air pollution values for the next 48 h (starting at 9 am each day) was developed. A novel broadcasting layer was incorporated in the model to introduce a spatial interpolation parameter into the deep-learning model training, and various LSTM-based deep-learning structures were used to support the end-to-end computation.

The proposed model, which combines ground observation data and WRF-CMAQ numerical models as the inputs, can forecast the air quality for any location within a region. In tests pertaining to China's Greater Bay Area (GBA) and surrounding regions, the proposed model outperformed the CMAQ model and an offline combination of the LSTM and SC methods in terms of the forecasting accuracy.

## 2 Method

### 2.1 Data

Ground observation data and WRF-CMAQ results from 2015 to 2021 in the GBA and surrounding regions (21.6–24.5 °N, 111.2–115.6 °E, referred to as the *target region* hereinafter), with a spatial resolution of 3 km, were extracted. Details of the model domain coverage and configuration of the parameterization schemes can be found in the work of Lu et al. (2015; 2018). The proposed model was built using the data from 2015 to 2020 (*training period*) and tested using the data from 2021 (*testing*

*period*) to ensure temporal generalizability.

The ground observation data of air pollutant concentrations from several ground monitoring stations distributed across the region were used to partially represent the spatial distribution of the pollutants. In the training period, the ground monitoring stations with at least 90% valid records (2015 to 2020) for both the target species, $PM_{2.5}$ and $O_3$, were selected as the *training target stations*. The same criterion was applied to select the *testing target stations* (2021) from the testing period. The ground

monitoring stations with at least 95% valid records for both the target species in both periods were selected as the *source stations* (denoted $S$), and the corresponding data were used as the ground truth for model training. Given these criteria, each source station was automatically a target station in both periods. As shown in Fig. 1, the criteria yielded 32 source stations, 90 training target stations, and 61 testing target stations. Twenty-one testing target stations that were neither source stations nor training target stations were used as the primary benchmark for quantitatively evaluating the results (referred to as *benchmark*

*stations* hereinafter; see Section 3). As the model did not encounter the data of these stations during training, satisfactory performances for these stations were expected to be indicative of spatial and temporal generalizability.

The WRF and CMAQ models can output the future weather situations and air pollutant concentrations, which represent valuable information for the deep-learning model. Therefore, the hourly WRF and CMAQ results for the forecast period at the locations of interest were input to the model. In other words, the WRF and CMAQ results for the training target stations were

used for the model training, and those for the testing target stations were used for the model testing. The WRF and CMAQ features are listed in List 1 in the supplementary materials.

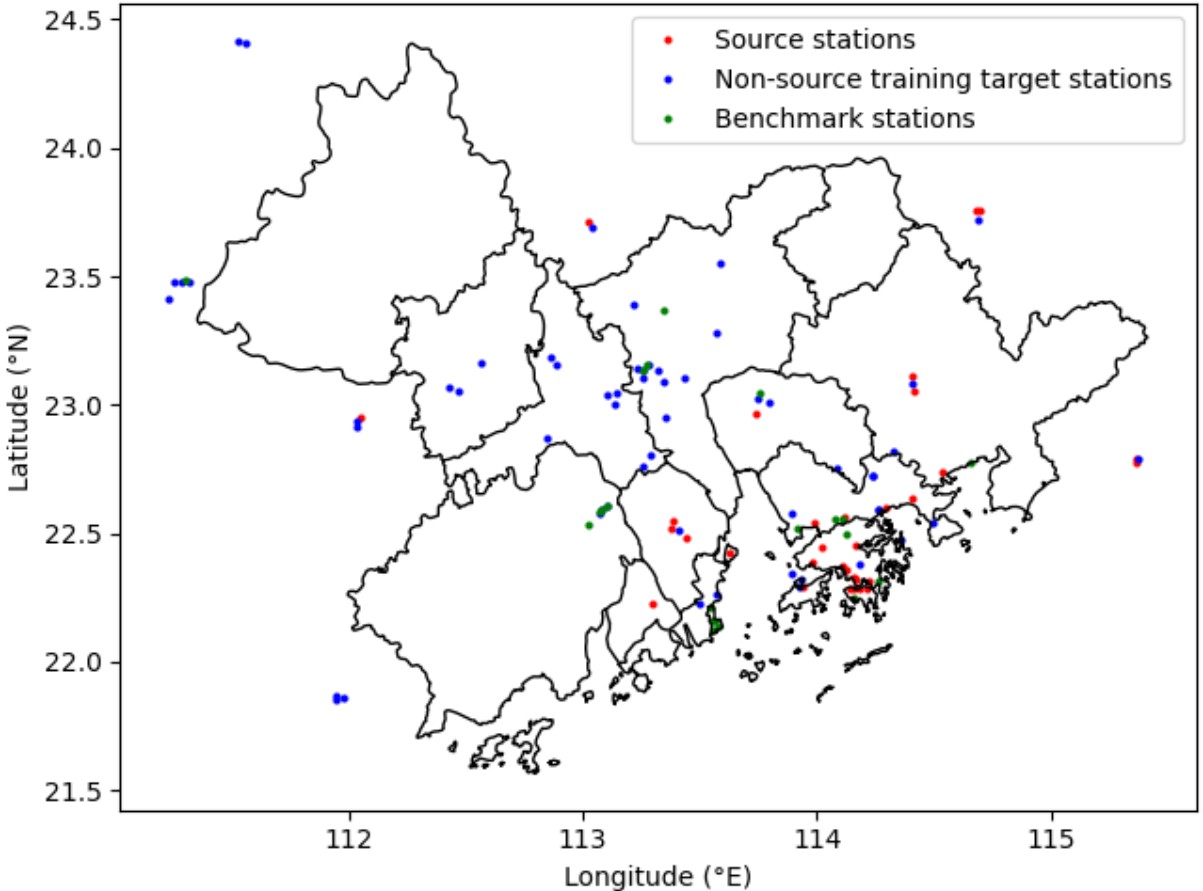

**Figure 1: Locations of the source and target stations (including the benchmark stations)**






For each day $d$, the proposed model took the following inputs:

1.  The hourly ground observation data at the source stations from 9 am, day $d-3$ to 8 am, day $d$ (both ends inclusive, 72 timesteps).
2.  The hourly WRF-CMAQ data at the locations of interest from 9 am, day $d$ to 8 am, day $d+2$ (48 timesteps).

The model then output the hourly forecast of PM$_{2.5}$ and O$_3$ concentrations at the locations of interest from 9 am, day $d$ to 8 am, day $d+2$ (48 timesteps).

## 2.2 LSTM encoder–decoders

The ground observation data at the source stations were processed using LSTM encoder–decoders, with one LSTM encoder–decoder associated with each source station. The LSTM encoder–decoder $f_s: X_s \to H$ of a source station $s$ is assumed to map

from the source-station-specific space $X_s$ of the past 72-h ground observation data (which may contain a different number of features for different source stations) to a homogeneous space $H$, representing the information related to the PM$_{2.5}$ and O$_3$ concentrations for the future 48 h of any location in the target region, derived from the past ground observations.

An LSTM encoder–decoder contains an encoder LSTM, a decoder LSTM, and a dense layer. First, the ground observation data for the past air pollutant concentrations and meteorological factors (denoted $\{\mathbf{x}^{(t)}\}_{t=1}^{T_{in}}$, where $T_{in} = 72$ h is the length of

the past observations) are input to the encoder LSTM to generate the encoding vector of the input time series, $\mathbf{h}$. Subsequently, $\mathbf{h}$ is passed to the decoder LSTM with $T_{out}$ timesteps, where $T_{out} = 48$ h is the length of the prediction. The hidden states of each timestep $\{\mathbf{h}^{(t)}\}_{t=1}^{T_{out}}$ are subsequently passed to a dense layer, activated by the rectified linear unit function (ReLU), where $\text{ReLU}(z) = \max(0, z)$, and applied to the output of the dense layer in an elementwise manner. The details of the encoder LSTM and decoder LSTM are presented in Texts S1 and S2 in the supplementary materials, respectively.

In this study, the LSTM encoder–decoder associated with each source station, regardless of the number of ground observation features, had an encoding dimension of 64. The output dimension of the dense layer was set as 64 (for each timestep). The structure of an LSTM encoder–decoder is illustrated in Fig. 2.

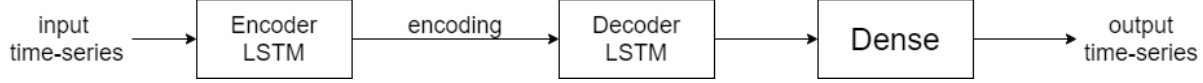

**Figure 2: LSTM encoder–decoder**

**2.3 Bidirectional LSTM**

Because several inputs and intermediate outputs (e.g., the WRF-CMAQ input at the locations of interest and outputs of the LSTM encoder–decoders) were in the same temporal space as that of the final output, bidirectional LSTMs were applied to extract the information embedded in these time series. A bidirectional LSTM contains two ordinary LSTM structures. When a time series $\{\mathbf{x}^{(t)}\}_{t=1}^{T}$ is input to a bidirectional LSTM, it is passed to the two LSTM layers in the ordinary and reversed



temporal orders, and the two hidden states of each timestep are concatenated as the output of the bidirectional LSTM. More

details regarding the bidirectional LSTM (as a variant of bidirectional RNNs) can be found in the work of Schuster and Paliwal

(1997).

## 2.4 Broadcasting layer

The SC method, which is based on numerical interpolation, has been introduced into the process of forecasting regional air

quality to address the asymmetry between the availability of information at a limited number of locations and the need to

predict the air quality for a complete region. For example, Sun et al. (2021) used the inverse distance weight to calibrate the

difference between the deep-learning forecast and CMAQ forecast. Ma et al. (2019) proposed a *geolayer* to filter the data used

for interpolation and combined this layer with LSTM-based models. However, in such offline combinations, the hidden

connection among different stations cannot be included in the deep-learning model building procedure. In addition, offline

numerical interpolation methods do not have degrees of freedom. Several methods of this type are not differentiable (e.g.,

nearest interpolation) or may incur numerical problems (e.g., inverse distance interpolation). Therefore, in order to better reveal

the spatial characteristics of the air pollutant concentration field, we introduced a novel *broadcasting layer* to enable the end-

to-end deep-learning model for regional air quality forecast.

In this framework, each ground observation station $s \in S$ is associated with a learnable weight decay parameter $\theta_s \geq 0$ (which

can be trained while building the deep-learning model). At any target location $t$, when the input $\{Y_s\}_{s \in S}$ is received from the

source stations, the output of the layer at location $t$ is computed as a weighted sum:

$$Y_t' = \sum_{s \in S} w_{s,t} Y_s, \tag{1}$$

with the weights calculated as

$$w_{s,t} = \frac{\exp\big(-\theta_s d(s,t)\big)}{\sum_{s' \in S} \exp\big(-\theta_{s'} d(s',t)\big)}, \tag{2}$$

where $d(\cdot, \cdot)$ denotes the distance between two locations, measured in kilometers. The computation of the weights is similar to

that implemented in the conventional SoftMax function. Therefore, the weights sum to 1 for each location $t$, and the numerical

problems that may occur during the differentiation of other forms (e.g., the inverse of the distance) are avoided. Because the

weighted sum preserves the dimensions, the output of the broadcasting layer (at a target location $t$) is a time series of 48

timesteps and 64 dimensions.

Model structure and training

Figure 3 shows the architecture of the proposed model. First, the ground observations of the source stations are passed through

the LSTM encoder–decoders, as described in Section 2.2, and broadcast to any location in the target region that requires the

forecast by using the broadcasting layer introduced in Section 2.4 (such a location is referred to as a *target location*). Then,

the WRF-CMAQ result for the target location and target hours (as a time series with a length of 48 and dimension of 10) is

passed through two bidirectional LSTM layers, both of which have an output dimension of 64. Next, the outputs of the





broadcasting layers and bidirectional LSTM layers are concatenated at each timestep, forming a time series with 48 timesteps and 128 dimensions. Finally, the combined time series is passed to another bidirectional LSTM layer with an output dimension of 64 and a time-distributed dense layer (i.e., a dense layer associated with each of the 48 timesteps) with an output dimension of 2, corresponding to the 48-h forecasts of the two air pollutant species.

In this study, the model was trained for 32 epochs using the ADAM optimizer (Kingma & Ba, 2014) by minimizing the mean absolute error (MAE) of prediction for all valid records, with a learning rate of $10^{-3}$ and batch size of 64. The following measures were introduced to prevent overfitting:

   1. A dropout layer (Srivastava et al., 2014) with a rate of 0.5 was applied before the dense layer in each LSTM encoder–decoder and before the final time-distributed dense layer.

2. A batch-normalization layer (Ioffe & Szegedy, 2015) was applied after each bidirectional LSTM layer (including the layers enclosed in the broadcasting layer). The WRF-CMAQ results and ground observations of the $PM_{2.5}$ and $O_3$ concentrations of the multiple training target stations were simultaneously fed to the model during each minibatch to attain a larger batch size for the batch-normalization layers to take effect.

This model is referred to as the *broadcasting model* hereinafter.

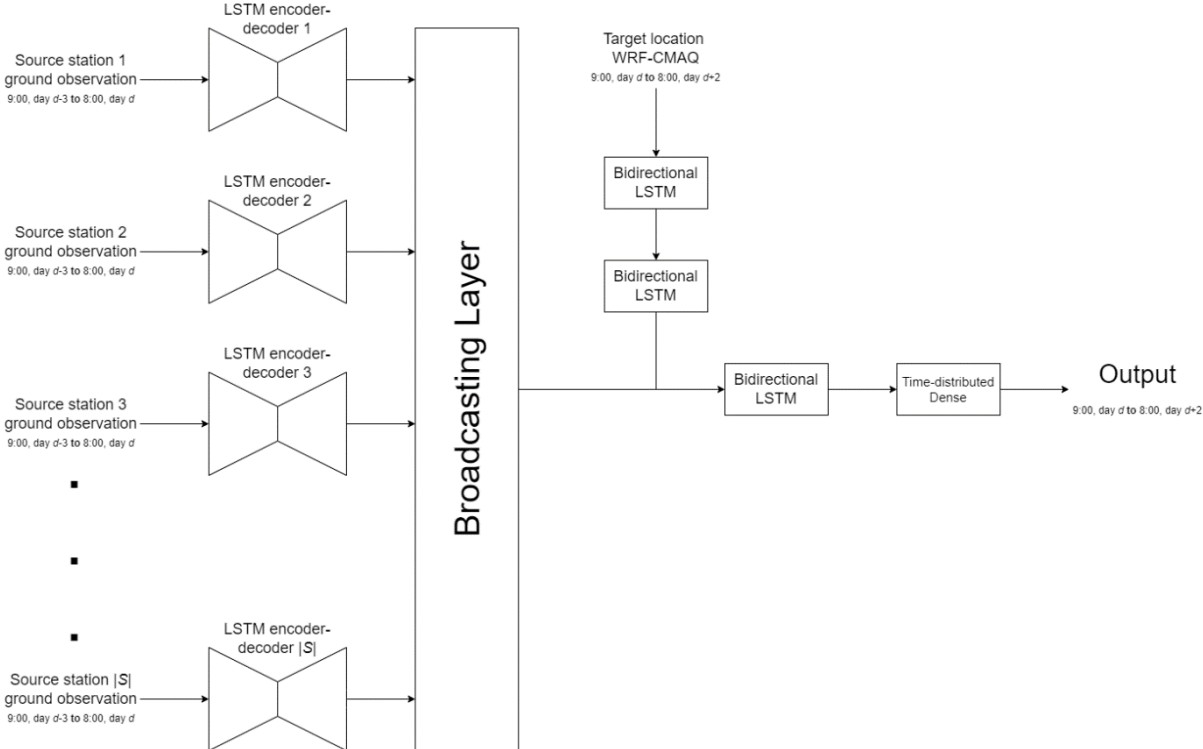


**Figure 3: LSTM-broadcasting model structure**





## 3 Results

The effectiveness of the broadcasting model was evaluated by comparing its results with the following two baselines:

1. The CMAQ model simulation.
2. The SC method introduced by Sun et al. (2021). Different interpolation methods [nearest neighbor (NN), inverse distance weighting (IDW), and kriging] were used to enhance the performance of the SC method on the test set.

The performance was evaluated using five metrics: mean bias error (MBE), mean absolute value (MAE), root mean square error (RMSE), symmetric mean absolute percentage error (SMAPE), and Pearson correlation coefficient (R). The formulas to determine these metrics are listed in Table S1 in the supplementary materials.

### 3.1 Overview

This subsection describes the performance evaluation of the broadcasting model against the baselines on the benchmark stations. Once the WRF-CMAQ forecast was available, the LSTM-broadcasting model required only several seconds to obtain the forecast for the next 48 h. Notably, the deep-learning-based structures of the SC were directly optimized to maximize the performance over the source stations. Therefore, the performance of SC on the target stations that were also source stations
could not be taken to represent its regional forecast performance.

The broadcasting model outperformed the baselines for all metrics for both $PM_{2.5}$ and $O_3$. Tables 1 and 2 summarize the performance values of the broadcasting model and baselines, temporally differentiated by two classes of time lag: 0–23 h and 24–47 h.

In terms of $PM_{2.5}$, the performance of all models in the first 24 h was superior to that in the second 24 h. According to the
MBE values, the CMAQ model was highly biased, and the SC only partially resolved this issue. The broadcasting model exhibited a significantly decreased bias for both the 24 h periods, and the forecast for the first 24 h was generally unbiased. Moreover, although the SC method outperformed the CMAQ model in terms of the MAE, RMSE, and SMAPE (especially with NN and IDW interpolations), it exhibited an inferior R value. In contrast, the broadcasting model exhibited an improved R value, indicating a decreased variance. Therefore, the overall error for the proposed model was considerably lower than those for the baselines. For example, the RMSE was 60% and 50% lower than those for the CMAQ and SC models, respectively,
and the improvement margins for the other metrics were significantly broader than those for the SC.

**Table 1: Overall performance values for the $PM_{2.5}$ forecast**

| Time lags | Model | MBE (µg/m³) | MAE (µg/m³) | RMSE (µg/m³) | SMAPE (%) | R |
|---|---|---|---|---|---|---|
| 0–23 h | CMAQ | 0.3678 | 12.62 | 21.86 | 59.14 | 0.5553 |
| | SC (NN) | 0.2855 | 11.60 | 18.64 | 62.82 | 0.4979 |
| | SC (IDW) | 0.3007 | 10.57 | 17.72 | 53.78 | 0.5146 |
| | SC (kriging) | 0.2040 | 10.05 | 16.51 | 55.80 | 0.5049 |





| | Broadcasting | **0.0163** | **6.455** | **9.157** | **41.24** | **0.7385** |
|---|---|---|---|---|---|---|
| | CMAQ | 0.3960 | 13.01 | 22.38 | 59.82 | 0.5487 |
| | SC (nearest) | 0.3503 | 12.56 | 19.51 | 65.25 | 0.4693 |
| 24–47 h | SC (IDW) | 0.3742 | 11.53 | 18.76 | 56.89 | 0.4923 |
| | SC (kriging) | 0.2689 | 10.93 | 17.21 | 58.75 | 0.4778 |
| | Broadcasting | **0.0973** | **7.237** | **9.759** | **45.06** | **0.7047** |

*The boldfaced values represent the highest performance for each period and metric.*

For $O_3$, similar to the case of $PM_{2.5}$, all models were more accurate in forecasting the $O_3$ concentrations in the first 24 h than
in the latter 24 h. However, unlike $PM_{2.5}$, the SC models outperformed the CMAQ model in terms of all metrics for $O_3$
forecasting, including R. The CMAQ was severely biased on both 24 h periods, although the SC solved the bias issue more
effectively than that in the case of $PM_{2.5}$. Notably, the broadcasting model calibrated the bias such that the model was generally
unbiased for both 24 h periods. In terms of the other metrics, the SC (especially with NN and IDW interpolations) exhibited
significant improvements over the CMAQ model (approximately 25% in terms of the MAE and RMSE and approximately 10%
in terms of the SMAPE and R). Nevertheless, the broadcasting model outperformed the SC, with improvements of nearly 10%
for all the metrics.

**Table 2: Overall performance values for the $O_3$ forecast**

| Time lags | Model | MBE (ppbv) | MAE (ppbv) | RMSE (ppbv) | SMAPE (%) | R |
|---|---|---|---|---|---|---|
| | CMAQ | 0.3346 | 16.09 | 21.06 | 60.05 | 0.6132 |
| | SC (NN) | -0.0654 | 12.38 | 16.67 | 58.14 | 0.6637 |
| 0–23 h | SC (IDW) | -0.0504 | 11.71 | 15.89 | 53.77 | 0.6835 |
| | SC (kriging) | -0.0507 | 11.53 | 15.62 | 54.16 | 0.6994 |
| | Broadcasting | **-0.0181** | **10.65** | **14.47** | **46.00** | **0.7411** |
| | CMAQ | 0.3305 | 16.33 | 21.22 | 61.08 | 0.6006 |
| | SC (NN) | -0.0687 | 12.66 | 17.11 | 58.74 | 0.6405 |
| 24–47 h | SC (IDW) | -0.0486 | 12.11 | 16.37 | 55.00 | 0.6594 |
| | SC (kriging) | -0.0520 | 11.88 | 16.10 | 55.11 | 0.6766 |
| | Broadcasting | **0.0174** | **11.19** | **15.22** | **47.52** | **0.7105** |

*The boldfaced values represent the highest performance for each period and metric.*

Figure 4 shows the hourly RMSE (representing the absolute error) and SMAPE (representing the relative error) of the forecasts
for the two pollutants. Owing to the daily-scale variations in the pollution levels, the RMSE and SMAPE trends were not
always consistent with one another, especially for $O_3$. In the case of $PM_{2.5}$, the performance of the baselines was unsatisfactory
at certain time lags (e.g., 11 h, 22 h, 35 h, and 46 h, corresponding to 8 pm and 7 am each day). In comparison, the broadcasting

model achieved satisfactory performance values over all time lags. In the case of $O_3$, the SC (especially with IDW and kriging interpolations) outperformed the CMAQ model for all metrics. The performance of the LSTM-broadcasting model at each

time lag was comparable to, if not better than, those of the baselines, and for most time lags, a significant margin of improvement was observed.

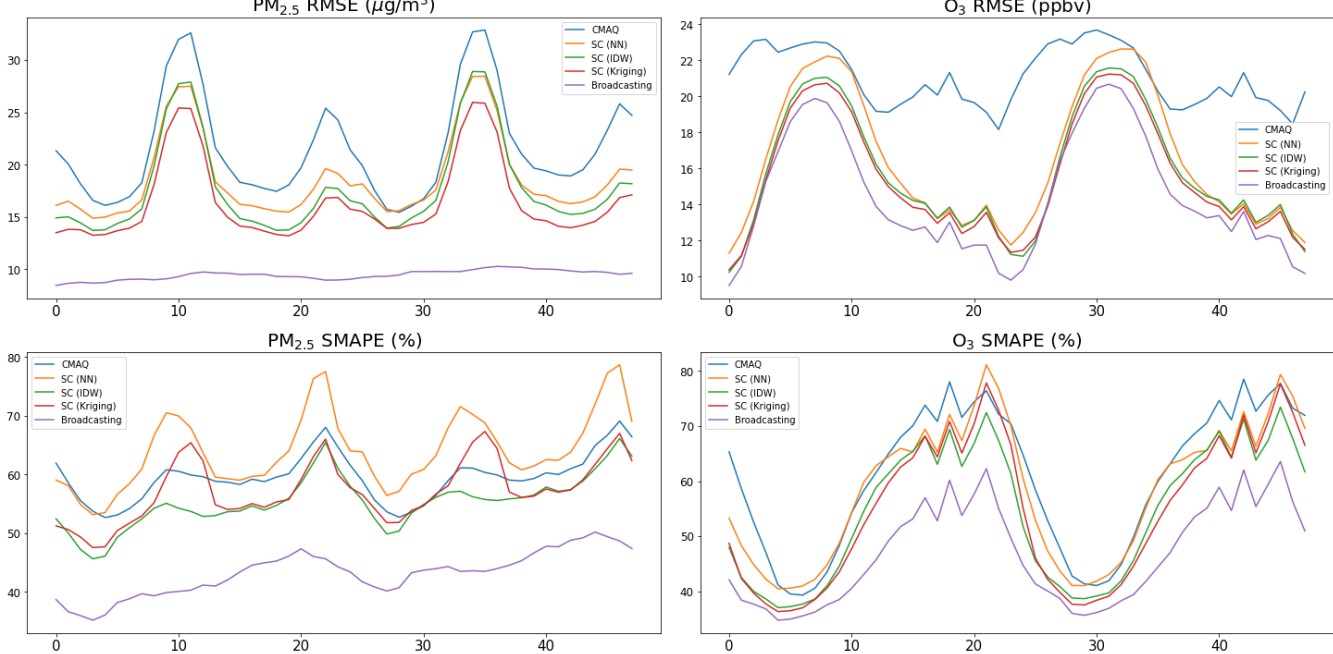

**Figure 4: Forecasting performance at each time lag**

Figure 5 compares the forecasts of the broadcasting model and baselines with the ground observation data in February, May,

August, and November 2021 (rows 1–4, respectively), considering the daily average over the benchmark stations. Consistent with the previous analyses, the forecast for the first 24 h was more accurate than that for the second 24 h. In the case of $PM_{2.5}$, the broadcasting model could better capture the trends of ground observations and was less vulnerable to systematic bias over long periods than the baselines. In the case of $O_3$, the SC model considerably outperformed CMAQ, and thus, the broadcasting model was not evidently more accurate than SC. Nevertheless, the results of the previous quantitative analysis demonstrated

the excellent capability of the broadcasting model in $O_3$ forecasting.







**Figure 5: Ground observations and forecasting data for February, May, August, and November 2021 (rows 1–4, respectively)**

### 3.2 Performance for different pollution levels

As described in Section 3.1, the proposed model achieved enhanced predictions compared with the baselines. The effectiveness

of the broadcasting model was further evaluated considering different levels of air pollution. The same 21 benchmark stations as those in the analysis described in Section 3.1 were used.

For each target pollutant, the daily averages of the ground observation values at the different stations were divided into four quartiles (Q1, Q2, Q3, and Q4, in increasing order), as indicated in Table 3:

**Table 3: Quartiles of PM$_{2.5}$ and O$_3$ concentrations**

| Pollutant | 25% | 50% | 75% |
|---|---|---|---|
| PM$_{2.5}$ (μg/m$^3$) | 9.833 | 15.68 | 24.10 |





| | | | |
|---|---|---|---|
| $O_3$ (ppbv) | 17.44 | 25.51 | 36.89 |


Figures 6 and 7 show the performance values (absolute and relative errors, indicated by the RMSE and SMAPE, respectively) of the broadcasting model and baselines for the four quartiles. For both the pollutants and all models, as the pollution levels increased, the absolute error increased and the relative error decreased. Similar to the overall performance trends, the different models were generally more accurate in the first 24 h in each quartile. However, the broadcasting model achieved significantly

improved $PM_{2.5}$ forecasts for all pollution levels. In particular, the RMSE of the broadcasting model was around 50% lower than that of the strongest baseline (SC with kriging interpolation) and was especially low at higher levels of pollution. A clear margin of improvement in the SMAPE was also observed at each quartile.

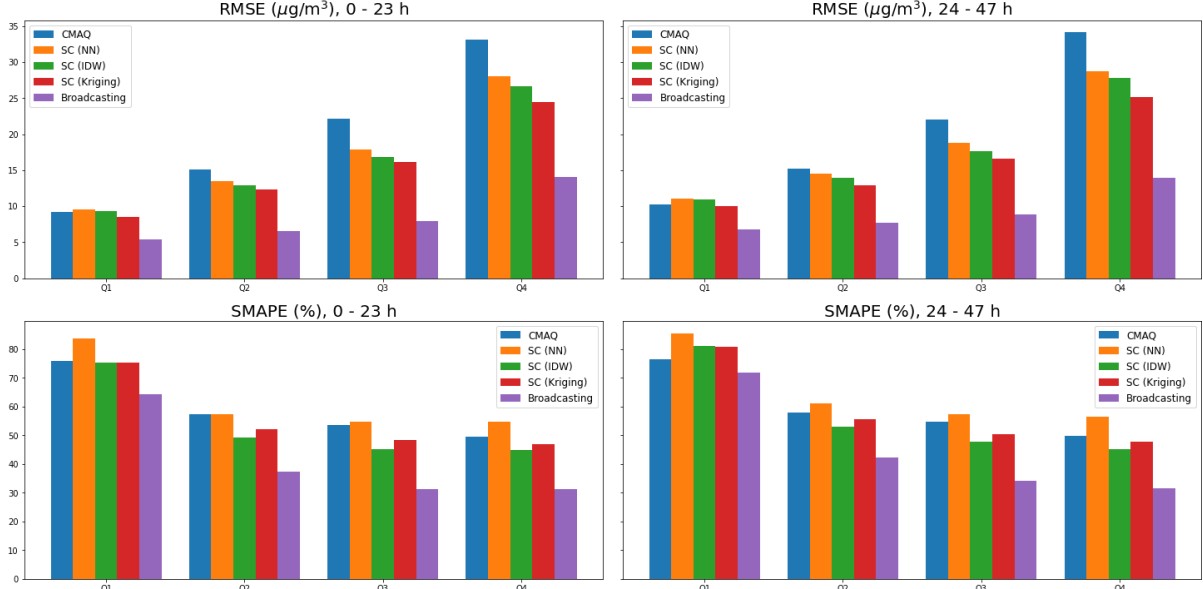

**Figure 6: PM₂.₅ forecast performances at each quartile**

The improvements of the broadcasting model for the $O_3$ forecasts were not as significant as those for the $PM_{2.5}$ forecasts. In certain cases (e.g., RMSE of the second 24 h for Q1), the broadcasting model did not outperform the SC (but still significantly outperformed the CMAQ model). However, in most cases, the broadcasting model still significantly outperformed the CMAQ and SC models, even given that SC (especially with IDW and kriging interpolations) already supersedes CMAQ by a large margin in many cases.

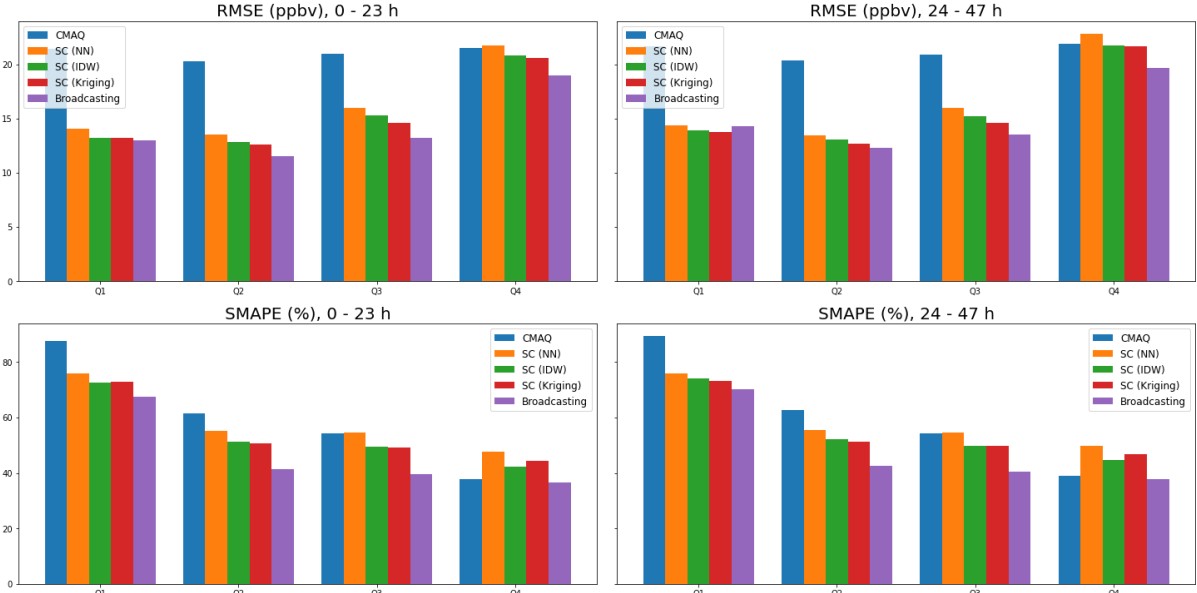

**Figure 7: O₃ forecast performances at each quartile**

In conclusion, in addition to the overall improvement in the forecast performance, as described in Section 3.1, the broadcasting model exhibit a satisfactory performance at different pollution levels. Therefore, the broadcasting model is robust against different scenarios and can be applied for high or low pollution levels.

Regional forecast

Figure 8 shows the regional forecast of the broadcasting model and baselines considering the monthly average of February 2021. The monthly average of the ground observations at the testing target stations is also shown for comparison. The regional forecasts for May, August, and November 2021 are presented in the supplementary materials (Figs. S1–S3).

The ground observations (dots) were typically inconsistent with the predictions (background) made by the CMAQ model. In other words, the CMAQ forecasts were generally inaccurate and biased (mainly positively), and could not accurately model the regional air pollution. The SC only partially resolved this issue, with occasional incompatibilities between the ground observations and forecasts. Moreover, owing to the mathematical characteristics of the different interpolation methods, the spatial distribution modeled using the SC framework was evidently unrealistic. For example, the SC forecast with NN interpolation exhibited apparent spatial discontinuities over several straight-line segments in the region, which is highly unrealistic.

In comparison, the ground observations and forecasts of the broadcasting model were consistent, which indicated that the broadcasting model could resolve the inaccuracies, especially the bias issue, encountered by the other models. Another key observation of the broadcasting model's prediction is that the spatial distribution simulated by the broadcasting model was smoother than those of the other models. However, the model that achieved the most realistic spatial smoothness cannot be identified from the given information owing to the lack of data in other regions. In fact, the smoothing effect may not align



with the fact that some cities (e.g., Guangzhou and Foshan) have higher emission levels than other locations in the target region. However, this inconsistency may also be attributable to the limited number of source stations in these cities (see Fig. 1). Nevertheless, in Hong Kong, in which the source stations are densely distributed, the broadcasting model successfully predicted a significantly lower pollution level, especially for PM$_{2.5}$.

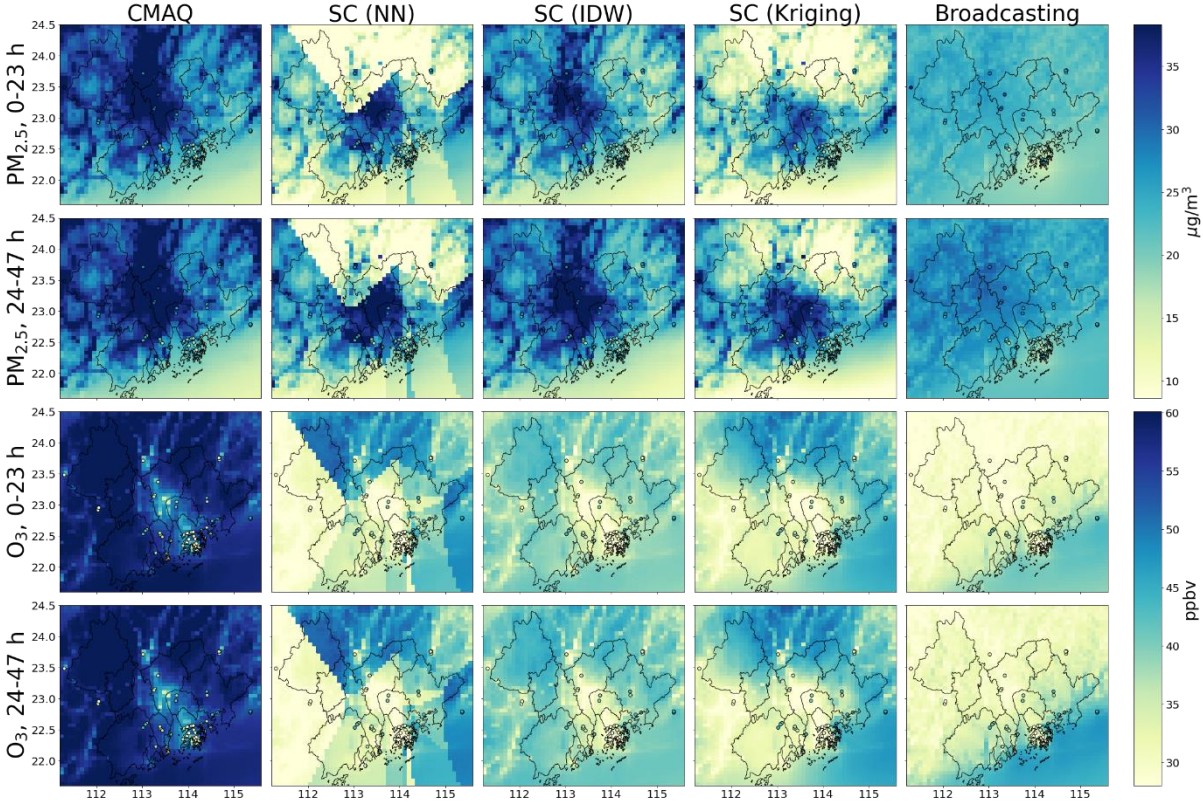


**Figure 8: Regional forecast results for February 2021**

## 4 Discussion

This paper proposes an end-to-end deep-learning model for regional air pollution forecast. The key structure enabling this feature is the broadcasting layer, which inputs the information extracted from the past ground observations at the discrete
source stations and projects it to any location in the target region as a weighted sum over all source stations. This layer can help overcome the geographical barrier and is a promising alternative to traditional customized SC methods that are typically based on inflexible and crude assumptions and result in exacerbated inaccuracies relative to the data of the ground monitoring stations. In addition, owing to the small number of parameters, the proposed model is unlikely to overfit spatially to the ground observation stations. The described structure can also be extended to regional air pollution forecasting or other deep-learning
tasks for regions for which information for only a limited number of locations is available. However, this study only assumed





that the impact of a source station decreases exponentially with the increase in the distance. Future work can be aimed at considering different patterns and factors other than the distance (e.g., terrains).

Moreover, the end-to-end deep-learning forecast does not incur a significant overhead, given that the ground observations and WRF-CMAQ results are available. As in Section 3.3, with GPU acceleration, the proposed model can obtain forecasts for
thousands or even tens of thousands of locations spread across the target region within several seconds. In contrast, if the interpolation methods (e.g., kriging) used by SC cannot be fully parallelized, the forecasting is associated with a prohibitive runtime, which decreases the applicability of such methods.

Instead of the conventional random splitting of the training and test sets, two disjoint periods were used for training and testing in this study. This design was motivated by the systematic long-term changes in the probability distributions of the pollutant
concentrations, which partially arise because of the implementation of emission reduction (Lu et al., 2020) and COVID-19 control measures (Fan et al., 2020), which must be considered when finetuning the model. If random splitting were applied, the trained and finetuned models would only be guaranteed to be valid on the data from 2015 to 2021 and may fail beyond this period.

In this study, a fixed set of source stations was considered, assuming that these stations would continuously output valid results
over the years. However, this design may result in loss of information. For example, if a ground observation station produced high-quality records between 2015 and 2018 but was later demolished, it was not selected as a source station. Moreover, this setting may cause some selected source stations to be invalidated in the future (e.g., if they are demolished in 2023). This problem could only be solved by considering an alternative setting in which the source stations are not selected statically but dynamically at each timestep (i.e., hour). However, this alternative setting would require the efficient management of the
varying source stations (even the variations in the number of these stations).

WRF-CMAQ simulation shows severe overestimations for both the $PM_{2.5}$ and $O_3$ forecasts, especially during 24-47 hours. The errors can be caused by several factors, such as the emission inventory, boundary and initial conditions, chemical and physical parameterization schemes and meteorological factors simulation. The emission inventory cannot always be up-to-date since substantial efforts are needed to compile a new set of regional emission inventory in high resolution. In addition, the scientific
community has not yet fully understood many of the chemical and physical mechanisms in the atmosphere, therefore, current state-of-the-art parameterization schemes still have a long way to be further improved. Besides devoting time and efforts to improving the performance of the prognostic model from the above mentioned perspectives, from this work, we can find that combining the observation data-driven skills (e.g., deep-learning methods) can work as a feasible and efficient option to make up the current deficiency inherent in 3D chemical transport model and thus improve the forecast performance.

**5 Conclusion**

Ground observations of recent hours can provide information regarding the most immediate meteorological and air pollution conditions. However, this information is typically available only for ground monitoring stations, and the absence of





information regarding the forecast period limits the accuracy of forecasts in the spatial and temporal dimensions. In this study, the parameters of spatial interpolation were incorporated into the training process by introducing a novel broadcasting layer.

This configuration could overcome the problems related to the offline SC methods and the spatial barrier, allowing information to be broadcast to all locations in the target region. Combined with the broadcasting layer, the end-to-end deep-learning model incorporated the ground observation and WRF-CMAQ results through different LSTM-based structures suitable for various formats of time-series data. The proposed model outperformed the existing models in terms of the $PM_{2.5}$ and $O_3$ forecasts. For the two pollutants, the absolute error (e.g., RMSE) of the proposed model was 55% and 30% lower than those of the CMAQ

model and 45% and 10% lower than those of the SC model. The relative error (SMAPE) of the proposed model was 30% and 20% lower than those of the CMAQ and 25% and 15% lower than those of the SC model. The proposed model structure can serve as a novel framework for regional air pollution forecasting. Specifically, this model can be applied to forecast the concentrations of $PM_{2.5}$, $O_3$, and other pollutants in different regions worldwide if adequate ground observations for the region are available and the numerical models (not necessarily WRF-CMAQ) can cover the target hours. The broadcasting layer may

also be further applied to a wide range of tasks that would otherwise require interpolation, thereby facilitating the development of end-to-end deep-learning models for these tasks. Considering the diverse natures of different tasks, ad hoc variations of the broadcasting layer may be designed to adapt to task-specific requirements.

**Code and data availability**

The ground air pollutant observation data was released by the China National Environmental Monitoring Centre and the Hong

Kong Environmental Protection Department. The ground observation data used in this study can be found at https://doi.org/10.5281/zenodo.6598377. The $PM_{2.5}$ and $O_3$ forecast results from different models and the setting of WRF-CMAQ model are available at https://doi.org/10.5281/zenodo.6833673. The WRF model v3.7 and CMAQ model v5.0.2 can be download from https://www2.mmm.ucar.edu/wrf/users/download/get_source.html and https://www.epa.gov/cmaq/access-cmaq-source-code. The official implementation of this work is at https://github.com/jvhs0706/regional-forecast-new, and the

deep learning model with the compatible format is available at https://doi.org/10.5281/zenodo.6827585.

**Author contribution**

HS, XL, and JF conceived the research. HS developed the model and performed the simulations, HS, XL, JF, and YC conducted the analysis. ZL, DY, and WC provided useful comments on the paper. HS prepared the manuscript, with contributions from all co-authors.



**Competing interests**

The authors declare that they have no conflict of interest.

**Acknowledgements**

This work was supported by the Guangzhou Scientific and Technological Planning Project (Project No. 202102021297), the National Natural Science Foundation of China (Project No. 42007203), the Research Grants Council of Hong Kong
Government (Project No.16305921), and the Environment and Conservation Fund (ECF 2020123).

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
