# Peer review of "Development of an LSTM-Broadcasting deep-learning framework for regional air pollution forecast improvement"

_Geoscientific Model Development, 2022_

## Author Comment (AC2)

We thank Anthony Fishwick for the constructive comments, which have helped us greatly improve the quality of our manuscript. Please see the point-to-point answer listed below; the corresponding changes have been reflected in our manuscript.

**Q1. The format.**

**1) Figure 1 no north arrow and scale bar.**

Thank you for the suggestion. We have added the north arrow and the scale bar in Figure 1.

**2) Tables 1&2 For the concentration values, please round to one or two decimal numbers because the models cannot predict so accurate till four decimal numbers. SC (NN) or SC(nearest)**

Thank you for pointing out the issues with Tables 1 and 2. We have corrected the typo and switched to 2 significant digits.

**3) Figures 2&3 The outline line was covered. Please replot it. Also, descriptions should be added to the figure caption because a figure is independent from the text. Also for the Tables.**

Thanks for the suggestion. We have followed the journal's formatting guidelines when making the tables, the plots, and other parts of the composition, making them easy to understand. The numbers of tables and figures have also been indicated in the text when they are referred to. We have added the description of Figure 2, please see Line 133-144. The description of Figure 3 can be found in Lines 178-186.

**4) Figure 5 The fonts are too small then it is hard to read. Please replot the figure.**

Thank you for pointing out the issue. We have replotted Figure 5 with a larger font size.

**5) Table 3 Here just three quartiles not four**

Thank you for pointing out the issue. However, Table 3 is meant to show the values of the 25%, 50% and 75% quantile, dividing 0% to 100% evenly into four quartiles. For example, in the first row, the 25%, 50%, and 75% quantiles are 9.833, 15.68, and 24.10, respectively, which means that the four quantiles are $[0, 9.833), [9.833, 15.68), [15.68, 24.10), [24.10, +\infty)$. We have changed Table 3 to the interval format to avoid confusion.

**Q2. There is totally no information on the computation cost. For the broadcasting model in this work, what is the computation burden for different cases? Please add one section on this, a trade-off between accuracy and computation cost should be considered rather than just the accuracy. This is important for the real application of air quality forecast (Zhang 2012, Atmospheric Environment; Lee 2020 Geosci. Model Dev. 1055-1073).**

Thank you for pointing out the issue. With GPU support, the Broadcasting model takes a few seconds to complete the regional forecast for each day (e.g., Section 3.3 and Figure 8). This constitutes the ignorable overhead compared with the time needed to obtain the WRF-CMAQ simulation results. For the SC models, depending on the nature of the interpolation methods, the regional forecast may take seconds if the interpolation can be parallelized (e.g., NN, IDW) or minutes otherwise (e.g., kriging). Therefore, the broadcasting model does not cause a significant computation burden, which satisfies the requirements for real applications (Zhang et al., 2012; Lee et al., 2020).

There is no significant trade-off relationship between the accuracy and the computational cost. We have added the following sentences about the efficiency of the Broadcasting model in lines 304-308:

"Moreover, the running time of the Broadcasting model is also reasonable. With the GPU (K80 in the Google Colab environment) support, it only takes several seconds to finish the computation for the regional forecasting of one day after the ground observation results and WRF-CMAQ data are available. Therefore, the Broadcasting model satisfies the efficiency requirements of real applications (Lee et al., 2020; Zhang et al., 2012). On the other hand, SC may take several seconds (NN and IDW) to about 3~5 minutes (Kriging), depending on whether interpolation methods can be fully parallelized."

Reference:

Lee, K., Yu, J., Lee, S., Park, M., Hong, H., Park, S. Y., ... & Song, C. H. (2020). Development of Korean Air Quality Prediction System version 1 (KAQPS v1) with focuses on practical issues. *Geoscientific Model Development*, *13*(3), 1055-1073.

Zhang, Y., Bocquet, M., Mallet, V., Seigneur, C., & Baklanov, A. (2012). Real-time air quality forecasting, part II: State of the science, current research needs, and future prospects. *Atmospheric environment*, *60*, 656-676.

**Q3. For the established model framework, why did the authors choose LSTM? The authors should mention the reasons/advantages.**

Thanks for the question. LSTM is one of the most successful and widely-used deep-learning structures for processing time series (Greff et al., 2016; Hochreiter & Schmidhuber, 1997; Karim et al., 2017; Siami-Namini et al., 2018). As a variant of RNN, it resolves the inherent problems of exploding and vanishing gradients, extending the effective forecasting horizon. Different variations of LSTM have been adopted in various scenarios of time series forecasting, depending on the diverse natures of the problems. We have chosen to use LSTM because our task heavily involves processing time series. As mentioned in lines 40 – 53, LSTM has been widely adopted in other literature on deep-learning-based air pollution prediction. In particular, as mentioned in Sections 2.2 and 2.3, LSTM encoder-decoders are suitable for the case when the output succeeds the input in temporal order, which applies to the ground observation input and the forecast; bidirectional LSTM, on the other hand, is suitable for processing the WRF-CMAQ input, which is in the same temporal space as the output. Therefore, we have combined these two variations of LSTM to construct our model.

We have added the following paragraph in Lines 323-329 to address our usage of LSTM:

"Also, our study has extensively exploited the power of LSTM in time-series-related deep learning tasks. LSTM is one of the most powerful deep-learning tools for time-series forecasting (Greff et al., 2016; Karim et al., 2017; Siami-Namini et al., 2018). As a variation of RNN, it resolves the inherent gradient explosion and vanishing problem, significantly extending the forecast horizon. By carefully examining the nature of different input and output components, we proposed combining two variations of LSTM – LSTM encoder-decoders and bi-directional LSTMs to construct the model, and achieved relatively good results. From this, we find that in complex time-series-related deep-learning tasks, careful and ad hoc analysis of the nature of the different input and output time series is needed to construct the most effective model and achieve higher accuracy."

Reference:

Greff, K., Srivastava, R. K., Koutník, J., Steunebrink, B. R., & Schmidhuber, J. (2016). LSTM: A search space odyssey. *IEEE transactions on neural networks and learning systems*, *28*(10), 2222-2232.

Hochreiter, S., & Schmidhuber, J. (1997). Long short-term memory. *Neural computation*, *9*(8), 1735-1780.

Karim, F., Majumdar, S., Darabi, H., & Chen, S. (2017). LSTM fully convolutional networks for time series classification. *IEEE access*, *6*, 1662-1669.

Siami-Namini, S., Tavakoli, N., & Namin, A. S. (2018, December). A comparison of ARIMA and LSTM in forecasting time series. In *2018 17th IEEE international conference on machine learning and applications (ICMLA)* (pp. 1394-1401). IEEE.

**Q4. I know that there are some similar previous studies (e.g., Sayeed 2021 Scientific Reports; Sayeed 2021 Atmospheric Environment; Lu 2021 Atmos. Pollut. Res. 101066; Lyu 2019 Environmental Science and Technology etc.). Therefore, the authors should clearly identify the scholarship of this work compared with previous studies and publications within your own research team.**

Thank you for raising the issue. As we have mentioned in our literature review of previous works on air pollution prediction using machine learning (lines 40 - 80) and the introduction of our contributions (lines 81 - 88), our study was the first to propose an **end-to-end deep learning model** for air pollution forecasting. In particular, the proposed broadcasting layer is a novel deep learning structure that can supersede the traditional spatial interpolation methods and achieve better forecasting accuracy in regional air pollution forecasting tasks.

In particular, our study also has significant novelty compared to the listed studies, which the reviewer suggested may be similar to ours. Sayeed et al. (2021a, 2021b) and Lu et al. (2021a) studies, although combining the ground observation data and CMAQ model with the deep learning techniques, focus on improving the forecasts **at the ground monitor stations**, which is different from our work which focuses on high-quality **regional forecasts**. On the other hand, although Lyu et al. (2019) have developed a deep-learning model for regional forecasts, the generalization from the scattered ground monitor stations to the whole region still depends on kriging interpolation, which is meant to be superseded by our study.

Similarly, as we have mentioned in the manuscript, this study also improves the regional air pollution forecasting of the LSTM-3D-VAR-CAMx (Lu et al., 2021b) and the LSTM-WRF-CMAQ (Sun et al., 2021) models, which were developed by our team previously. Experiments (Section 3) have shown that this study has significant advantages over our previous studies in both accuracy and efficiency.

To further clarify the novelty of this study, we have added the following sentences in lines 63 – 67:

"Sayeed et al. (2021a, 2021b) and Lu et al. (2021a) improved the accuracy and horizon CMAQ forecast by ground observations using deep learning techniques, but the improvements were still limited to the ground monitor stations rather than the whole region. On the other hand, Lyu et al. (2019) developed an ensemble model that combines the chemical transport models and the ground observations, but the regional forecast still depends on the traditional kriging method."

Reference:

Lu, H., Xie, M., Liu, X., Liu, B., Jiang, M., Gao, Y., & Zhao, X. (2021a). Adjusting prediction of ozone concentration based on CMAQ model and machine learning methods in Sichuan-Chongqing region, China. *Atmospheric Pollution Research*, *12*(6), 101066.

Lu, X., Sha, Y. H., Li, Z., Huang, Y., Chen, W., Chen, D., ... & Fung, J. C. (2021b). Development and application of a hybrid long-short term memory–three dimensional variational technique for the improvement of PM2.5 forecasting. Science of The Total Environment, 770, 144221.

Lyu, B., Hu, Y., Zhang, W., Du, Y., Luo, B., Sun, X., ... & Russell, A. G. (2019). Fusion method combining ground-level observations with chemical transport model

predictions using an ensemble deep learning framework: application in China to estimate spatiotemporally-resolved PM2.5 exposure fields in 2014–2017. Environmental science & technology, 53(13), 7306-7315.

Sayeed, A., Choi, Y., Eslami, E., Jung, J., Lops, Y., Salman, A. K., ... & Choi, M. H. (2021a). A novel CMAQ-CNN hybrid model to forecast hourly surface-ozone concentrations 14 days in advance. *Scientific reports*, *11*(1), 1-8.

Sayeed, A., Lops, Y., Choi, Y., Jung, J., & Salman, A. K. (2021b). Bias correcting and extending the PM forecast by CMAQ up to 7 days using deep convolutional neural networks. *Atmospheric Environment*, *253*, 118376.

Sun, H., Fung, J. C., Chen, Y., Chen, W., Li, Z., Huang, Y., ... & Lu, X. (2021). Improvement of PM2.5 and O3 forecasting by integration of 3D numerical simulation with deep learning techniques. Sustainable Cities and Society, 103372.

---

## Author Comment (AC3)

**In this paper, the authors introduce an air pollutant (ozone and PM2.5) forecasting model system which based on the deep-learning method. With the implementation of ground observations and the outputs of 3D chemical transport model, this model can forecast more accurate concentrations of ozone and PM2.5. Further, this model system can extend the prediction of air pollutants from individual station to a regional forecasting by considering the temporal characteristics of the time series and spatial relationships among different stations. By comparing the result with the observations, the results of this model show a more accurate and reasonable distributions of ozone and PM2.5, which indicates that this model system can work as a feasible and efficient option to improve current forecast performance. I think the authors did an interesting work. And this study is within the scope of GMD journal. Some problems need to be solved before it can be published.**

[**Response**]: We want to express our sincere thanks to Anonymous Referee #1 for acknowledging the significance of our study. Moreover, the valuable comments from the referee have also offered us great help in improving the quality of the manuscript. Please refer to the following point-to-point response to the comments. The corresponding changes have been reflected in the revised version of the paper.

**Comments:**

**"The ground monitoring stations with at least 90% valid records" (Line 92~93) and "the ground monitoring stations with at least 95% valid records ……were selected as the source stations" (Line 95~96). How are these threshold values determined? Please describe detailed information about these threshold values determination methods.**

[**Reponse**]: Thank you for your question. Generally speaking, the threshold values have been determined adaptively from the nature of the dataset, based on our experience from previous studies of similar nature (Lu et al., 2021; Sun et al., 2021).

The completeness of data is critical to the quality of our study. For many stations, long periods of continuous invalid records exist, which indicates that the measurement stations have undergone a systematic failure or closure. These periods may last for months or even years, so applying interpolation methods to fill these gaps is improper. Otherwise, the interpolated data may harm the training process, especially as our model uses LSTM to process the time-series data. Moreover, large fault rates indicate that the ground monitoring stations may be unreliable, in the sense that even those data that are not invalid are still untrustworthy, which may harm

the model's performance profoundly. Therefore, we set the threshold values relatively high, 90% for the target stations, and even higher (95%) for the source stations, to filter out the stations that may harm the training.

Moreover, the threshold for the source stations is higher than that for target stations because the source stations are relatively more important than the target stations: the source stations provide data as part of the input to the Broadcasting model and are a part of the model. In contrast, the target stations only provide the ground truth values as a reference during the training and testing procedures (please see our response to question 2 for details). Theoretically, the trained model could be more reliable if all the source stations could reach a 100% valid rate. However, in practice, the 100% valid rate is unrealistic, and we set the value to 95% to keep an adequate number of stations. In the ideal case, if all source stations have 100% valid records, the model's performance could be further improved.

Despite the integrity of the data, we also have to consider the consequence of filtering out too many stations. Figures 1 and 2 show the threshold values for each source station and training target station to be filtered out. If the threshold for source stations (95%) is set higher (e.g., 97%), many source stations will be filtered out, and the prediction in a vast portion of the target region would be rendered questionable. Similarly, if the threshold value for target stations (90%) is set higher (e.g., 95%), the easternmost cluster of stations (around 23.5°N, 111.2°E) would be filtered out, which will cause the model to be not well-trained for the surrounding areas.

[Figure]

Figure 1: The threshold values for each source station to be filtered out.

[Figure]

Figure 2: The threshold values for each training target station to be filtered out.

To further clarify the selection process of the source and target stations, we have added the following sentences in lines 107-109:

"Note that the threshold values of the selection criterion are determined adaptively from the nature of the dataset. The values were set relatively high such that the quality of the data could be guaranteed. However, to ensure that an adequate amount of stations are selected to represent different areas of the target region, the threshold values could not be set too close to 100%."

**2. The data of source stations is also used for model training. What is the difference between the data of source stations and which of training stations?**

**[Response]:** Thank you for the question. There is a primary difference between the source stations and the training target stations (referred to as *training stations* by the reviewer). The source stations serve as part of the **input** to the model **throughout the whole pipeline**, while the training target stations only serve as the **output during the training phase**.

As shown in Figure 3 of the preprint, each source station is associated with an LSTM encoder-decoder. Therefore, when making a forecast (for the target region), the model expects the input from all the source stations. That is to say, the set of source stations is **fixed** and **part of the Broadcasting model**.

On the other hand, the training target stations only play a role in the training phase. The ground observation values of the two target pollutants in the training target stations ($PM_{2.5}$ and $O_3$) for the future two days are used as reference in training. After the training, the training target stations are no longer relevant, and the model can forecast for **any place in the target region**, rather than limited to the training target stations or testing target stations.

[Figure]

*Figure 3: LSTM-broadcasting model structure (also Figure 3 in the manuscript)*

**3. Line 114. I am confused that why the authors use the WRF-CMAQ data with the next two days to training the model?**

**[Response]:** Thank you for the question. Recall that the WRF-CMAQ model can also used to do the forecast. Therefore, its result is also available for the future two days. Concretely, following the setting in lines 119-126, when making the forecast for 9 am, day $d$ to 8 am, day $d + 2$, the WRF-CMAQ results for these 48 hours will be available.

As mentioned in lines 110-114, the WRF-CMAQ results for the coming 48 h carry valuable information about this period's weather and air pollution conditions. Therefore, including it in the input can significantly improve forecasting accuracy, as verified by previous studies (Sun et al., 2021; Lu et al., 2020).

We have added the following sentence in lines 122-124 to enhance the clarity:

"Note that the WRF-CMAQ model can also work as a forecasting model, and therefore these data are available and can be used before the beginning of the forecast."

**4. In this model system, large quantities of observations are used for training model. Please discuss the impact of the number of training stations and source stations on the model performance?**

[**Response**]: Thank you for the question. To answer this question, we will first focus on the Broadcasting layer, the main deep-learning structure connecting the source stations and the training target stations.

The development Broadcasting layer is based on Tobler's first law of geography, which states that *"everything is related to everything else, but near things are more related than distant things."* The design of the Broadcasting layer, as described in Section 2.4, has the property that the impact of a source station on a target location decreases as its distance increases, which follows the law. The Broadcasting layer is a deep-learning alternative to the traditional spatial interpolation methods (e.g., inverse distance weighted, kriging), which also follow the law. Therefore, the Broadcasting layer can model the spatial distribution of air pollution caused by the relative locations and becomes the first deep learning structure to model this law to the best of our knowledge.

However, the spatial distribution of $PM_{2.5}$ and $O_3$ may be complex and depends on multiple factors (e.g., the meteorological field, terrain, etc.). Therefore, a substantial density of source stations and training target stations is necessary for modelling the air pollution in different areas of the target region:

If an area in the target region has few to no source stations, then the model would not gain sufficient information from the past meteorological conditions and air pollution in this area. Therefore, the forecasting accuracy may be impacted in areas with lower source station density. On the other hand, if the training target stations in an area are low in density, the model will not be trained to perform well in that area. This is because the training mechanism will put a larger weight on the areas with denser distributions of training target stations, and the other regions may be "ignored" by the training algorithm. The impact of the number of source stations and training target stations has been shown in Figure 8 and discussed in Lines 293-301.

Therefore, a relatively uniform distribution source and training target stations may also improve the model's performance. As a result, other selection strategies may be developed considering this factor in future works to improve the regional forecasting accuracy further.

To make our discussion more comprehensive, we have added the following content in lines 348-352:

"In our setting, the source and training target stations play an essential role in the model's accuracy. The forecast quality generally increases as the number of source and training target stations increases. Therefore, the model's performance has been uneven across different areas of the target region: for example, as shown in 3.3, the performance in Hong Kong is generally better than that in other regions. In future works, other selection criteria of source stations and training target stations, in place of those introduced in Section 2.1, may be developed to resolve this issue."

Reference:

Lu, X., Sha, Y. H., Li, Z., Huang, Y., Chen, W., Chen, D., ... & Fung, J. C. (2021). Development and application of a hybrid long-short term memory–three dimensional variational technique for the improvement of PM2.5 forecasting. *Science of The Total Environment*, *770*, 144221.

Sun, H., Fung, J. C., Chen, Y., Chen, W., Li, Z., Huang, Y., ... & Lu, X. (2021). Improvement of PM2.5 and O3 forecasting by integration of 3D numerical simulation with deep learning techniques. *Sustainable Cities and Society*, 103372.

---

## Author Comment (AC4)

**This manuscript (gmd-2022-164) proposed a new approach to refine air pollutant forecasting using deep-learning techniques. Based on the LSTM technique, a novel broadcasting layer was introduced to provide results with spatial coverage. This study not only extends the LSTM-3D-VAR method (Lu et al., 2021), but also demonstrated an alternative to achieve spatial coverage. The ability to extend single site results to multiple locations is useful for real-world applications and of great interest to the atmospheric community. The performance of the new integrated modal was carefully evaluated, and the results are promising. The reviewer thinks the manuscript can be published on GMD after addressing the following issues.**

**[Response]:** We want to express our sincere thanks to Anonymous Referee #2 for acknowledging the significance of our study. Moreover, the valuable comments from the referee have also offered us great help in improving the quality of the manuscript. Please refer to the following point-to-point response to the comments. The corresponding changes have been reflected in the revised version of the paper.

**1. Line 144. Please provide the full name of "The SC method".**

**[Response]:** We thank the referee for pointing out this issue. The full name of the SC method, **spatial correction**, has been added in the corresponding location as specified by the reviewer. Please see Line 156.

**2. The authors mentioned that the LSTM-3D-VAR model (Lu et al., 2021) required substantial computation power. How about the computational efficiency of LSTM-Broadcasting compared with LSTM-3D-VAR? Does LSTM-Broadcasting consume less computation power than the LSTM-3D-VAR model? If possible, please include the direct comparison of the two models developed by the authors.**

**[Response]**: Thank you for the question. LSTM-3D-VAR-CAMx will cost about 90 minutes when the numerical forecast was performed in a cluster machine with 40 cores and 128GB memory, after the ground observation and the numerical models (WRF-CAMx) simulation results are available. For this LSTM-Broadcasting deep-learning framework, with the GPU (Google Colab K80) support, it only took several seconds to finish the computation when making the forecast for each day after receiving the ground observation data and WRF-CMAQ results. Therefore, the LSTM-Broadcasting model does not constitute any significant computational overhead and is much more efficient when compared to the LSTM-3D-VAR-

CAMx scheme developed by the authors before. We have added below sentences to Lines 304-311 in the manuscript:

"Moreover, the running time of the Broadcasting model is also reasonable. With the GPU (K80 in the Google Colab environment) support, it only takes several seconds to finish the computation for the regional forecast of one day after the ground observation results and WRF-CMAQ data are available. Therefore, the Broadcasting model satisfies the efficiency requirements of real applications (Lee et al., 2020; Zhang et al., 2012). On the other hand, SC may take several seconds (NN and IDW) to about 3~5 minutes (Kriging), depending on whether interpolation methods can be fully parallelized. By contrast, the LSTM-3D-VAR-CAMx will cost about 90 minutes (tested on a cluster machine with 40 cores and 128GB of memory) given the ground observation and WRF-CAMx results as input, which may render the approach infeasible when instant forecasts are needed."

**3. Line 281. "As in Section 3.3, with GPU acceleration…". Please check, section 3.3 cannot be found in the manuscript.**

**[Response]**: We thank the referee for pointing out this issue. We have corrected the formatting error, and please see line 282 in the manuscript.